# Cerebellar Agenesis and Bilateral Polimicrogyria Associated with Rare Variants of CUB and Sushi Multiple Domains 1 Gene (CSMD1): A Longitudinal Neuropsychological and Neuroradiological Case Study

**DOI:** 10.3390/ijerph19031224

**Published:** 2022-01-22

**Authors:** Floriana Costanzo, Ginevra Zanni, Elisa Fucà, Margherita Di Paola, Sabina Barresi, Lorena Travaglini, Giovanna Stefania Colafati, Antonio Gambardella, Emanuele Bellacchio, Enrico Bertini, Deny Menghini, Stefano Vicari

**Affiliations:** 1Child and Adolescent Neuropsychiatry Unit, Department of Neurosciences, Bambino Gesù Children’s Hospital IRCCS, Via Ferdinando Baldelli 41, I-00146 Rome, Italy; floriana.costanzo@opbg.net (F.C.); elisa.fuca@opbg.net (E.F.); stefano.vicari@opbg.net (S.V.); 2Unit of Neuromuscular and Neurodegenerative Disorders, Department of Neurosciences, Bambino Gesù Children’s Hospital, IRCCS, Viale di San Paolo 15, I-00146 Rome, Italy; ginevra.zanni@opbg.net (G.Z.); lorena.travaglini@opbg.net (L.T.); enricosilvio.bertini@opbg.net (E.B.); 3Department of Clinical and Behavioral Neurology, IRCCS Santa Lucia Foundation, Via Ardeatina 306, I-00179 Rome, Italy; m.dipaola@hsantalucia.it; 4Department of Mental Health, King Faisal Specialist Hospital & Research Center, Riyadh 12713, Saudi Arabia; 5Pathology Unit, Department of Laboratories, Bambino Gesù Children’s Hospital, IRCCS, Viale di San Paolo 15, I-00146 Rome, Italy; sabina.barresi@opbg.net; 6Oncological Neuroradiology Unit, Department of Imaging, Bambino Gesù Children’s Hospital, IRCCS, Piazza Sant’Onofrio 4, I-00100 Rome, Italy; gstefania.colafati@opbg.net; 7Institute of Neurology, University Magna Græcia, I-88100 Catanzaro, Italy; a.gambardella@unicz.it; 8Institute of Molecular Bioimaging and Physiology, National Research Council, I-88100 Catanzaro, Italy; 9Genetics and Rare Diseases Research Division, Bambino Gesù Children’s Hospital, Viale di San Paolo 15, I-00146 Rome, Italy; emanuele.bellacchio@gmail.com; 10Department of Life Science and Public Health, Catholic University of the Sacred Heart, Largo Agostino Gemelli 1, I-00168 Rome, Italy

**Keywords:** agenesis, cerebellar ataxia, polymicrogyria, CUB and Sushi multiple domains 1 gene

## Abstract

Cerebellar agenesis is an extremely rare condition characterized by a near complete absence of the cerebellum. The pathogenesis and molecular basis remain mostly unknown. We report the neuroradiological, molecular, neuropsychological and behavioral characterization of a 5-year-old girl, with cerebellar agenesis associated with parietal and peri-Sylvian polymicrogyria, followed-up for 10 years at four time points. Whole exome sequencing identified two rare variants in *CSMD1*, a gene associated with neurocognitive and psychiatric alterations. Mild intellectual impairment, cerebellar ataxia and deficits in language, memory and executive functions, with relatively preserved adaptive and psychopathological domains, were initially showed. Phonological awareness and verbal memory declined at 11 years of age, and social and anxiety problems emerged. Adaptive and psychopathological characteristics dramatically worsened at 15 years. In summary, the developmental clinical outcome showed impairment in multiple cognitive functions in childhood, with a progressive decline in cognitive and adaptive abilities and the emergence of psychopathological symptoms in adolescence. The observed phenotype could be the result of a complex interplay between cerebellar abnormality, brain malformation and the relations with *CSMD1* variants. These findings may provide insights into the developmental clinical outcomes of a co-occurrence between rare brain malformation and rare genetic variants associated to neurodevelopmental disorders.

## 1. Introduction

Cerebellar ontogenesis is orchestrated by a complex interaction of cell-autonomous programs and environmental factors (for a review, see [1]). The development of the cerebellum extends for a long period, starting around the ninth gestational week and continuing beyond birth; this protracted developmental timeline makes the cerebellum particularly prone to malformations and disruptions, sometimes associated with intervening epigenetic factors [1,2,3]. Among these, total or partial cerebellar agenesis is a rare condition [4,5,6], usually characterized by the presence of small portions of cerebellar tissue, i.e., remnants of middle cerebellar peduncles, anterior vermal lobules and/or flocculi [7,8,9]. Cerebellar agenesis has been found associated with gene mutations, such as *PTF1A* [10], or as a secondary disorder following conditions such as prematurity [9] or pre/perinatal hemorrhage [7].

Given the high mortality rate, few and inconsistent behavioral data are available from living patients with cerebellar agenesis [11]. The paucity and inconsistency of available data gave rise to an ongoing debate about the degree of cerebellar agenesis impact on individual functioning.

Clinically, cerebellar agenesis is often characterized by ataxia and impaired movement [9,12] with different outcomes ranging from early death to variable degrees of motor dysfunctions [9,13]. In the last two decades, great attention has also been paid to the non-motor facets of cerebellar disruptions, including cerebellar agenesis [14,15]. Despite the growing amount of evidences on neuropsychological and behavioral correlates of cerebellar agenesis, a high heterogeneity of findings persists.

The idea that cerebellar agenesis is entirely symptom free—thus allowing a “normal life”—has largely spread out over time. Starting from one of the first cases reported in literature, dating back to 1940 [16], a growing amount of case descriptions have presented a mild impaired picture associated with cerebellar agenesis. The cases of a 6-year-old girl and of a 58-year-old woman with near total cerebellar agenesis without intellectual disability or neurobehavioral symptoms have been described [17,18]. Moreover, the case of a 22-year-old man [19] with only motor impairment, for instance, ataxia, associated with typical neuropsychological development, was observed. However, Ashraf and colleagues [20] reported a case in which cerebellar agenesis was not associated with motor impairment, dysarthria nor nystagmus but only with learning difficulties. Yu and colleagues [21] described the case of a 24-year-old woman with complete cerebellar agenesis exhibiting only mild to moderate signs of motor impairment and cerebellar dysarthria. More recently, Wu and colleagues [22] reported the case of a 26-year-old patient with complete primary cerebellar agenesis, exhibiting mild to moderate motor impairment associated with impairment in associative motor learning. In other cases, mild intellectual disability with language difficulties but preserved abilities, such as reading or riding a bicycle, as well as adequate levels of affective behavior were detected. An explanation of these results is the hypothesis that the extra-cerebellar motor system can compensate for lost cerebellar motor functions [23].

On the other side, several studies reported histories of patients with multiple and severe deficits associated with cerebellar agenesis such as executive functions, behavioral and/or neuropsychological alterations, multisensory integration associated with ataxic gait and oculomotor disorders [13,24,25,26]. Chedda, Sherman and Schmahmann [24] reported two cases of children with near-complete agenesis with gross and fine motor deficits, such us oral motor apraxia, impaired saccades and vestibulo-ocular reflex cancellation, clumsiness and mild ataxia. Behavioral characteristics included autistic-like stereotypical performance, obsessive rituals and difficulty in understanding social cues. The most important neuropsychological deficits affected executive functions (perseveration, disinhibition, abstract reasoning, working memory and verbal fluency), visual-spatial abilities (perceptual organization, visual-spatial copying and recall) and expressive language (delay). The severity and range of the motor, cognitive and psychiatric impairments were also related to the extension of the agenesis.

Timmann and colleagues [13] reported the case of a 59-year-old patient, with almost total cerebellar agenesis, with a number of oculomotor, speech and gait control deficits as well as developmental delay and learning deficits. Mild to moderate deficits in IQ and reduced planning abilities, visual-spatial abilities, visual memory and attention or deficits in speech comprehension, verbal learning and declarative memory, were also described [26,27,28]. More recently, it has been reported the case of a 48 years old man with cerebellar agenesis, exhibiting deficits in executive functions (planning, flexibility and focused attention) and multisensory integration, associated with ataxic gait and oculomotor disorders [25].

Although documented detailed neurologic, neuropsychiatric and neuroimaging findings in living patients with total cerebellar agenesis are limited, the patients described present with a variety of developmental motor, cognitive and behavioral abnormalities.

Considering neuroanatomical features, cerebellar agenesis is often associated with different types of cortical abnormalities, such as callosal hypoplasia, abnormal basal ganglia and polymicrogyria, a malformation secondary to abnormal post-migrational development [29,30]. Such conditions have been found to worsen the clinical outcome of isolated cerebellar agenesis in children [31]. Mutations in the α- and β-tubulin genes have been identified in polymicrogyria with additional cerebellar malformations [30,32,33,34], however, the relations between genes mutations, neuroanatomical features and neuropsychological characteristics of the associated malformations are still only partially understood.

The heterogeneity of the genetic and neuroanatomical correlates associated with cerebellar agenesis is probably at the origin of the contrasting results. Moreover, the lack of longitudinal studies investigating developmental trajectories of cerebellar agenesis contributes to the unclear outcome profile. Given that contrasting results about the correlates and the outcomes and of cerebellar agenesis still exist, longitudinal and detailed descriptions of this rare condition are strongly needed.

Here, we report the case of a girl with ataxia and subtotal cerebellar agenesis and cortical abnormalities. We performed genetic and neuroradiological examinations and characterized, at different time points, her neuropsychological phenotype by an extensive battery of tests covering a large range of neuropsychological domains, including language, memory, executive functions, perceptual and visual-spatial abilities. The girl’s adaptive, psychopathological and behavioral characteristics were also evaluated.

### Clinical Case

The first time we tested the girl, she was 5 years and 8 months old. She was born at 36 weeks of gestation by caesarean delivery. Her parents were unrelated and did not reported a family history of neuropsychiatric disorders. She had no siblings. The socioeconomic status of the family was middle class. At birth, the girl had a bilateral club foot; she weighed 2.150 Kg, was 44 cm long and her occipito-frontal circumference was 33 cm. She was the child of a single pregnancy complicated by maternal hypertension and diabetes and threat of miscarriage in the first trimester. At birth, she needed resuscitation because the cord was around her neck, but no mechanical ventilation was needed. During the follow-up, club feet were corrected surgically and she was able to walk at age 3 years. Although the first words were articulated at 6–7 months of age, her subsequent language development was delayed. She gained sphincter control at age 3 years. At 21 months she developed seizures that partially responded to Valproate 100 mg/day. After the first seizure, she was submitted to a Computerized Tomography, which revealed cerebellar agenesis.

At the first clinical observation, the girl exhibited typical signs of cerebellar ataxia, with unsteady gait, dysmetria, dysarthria, dysdiadochokinesia on the left and mild strabismus. Moreover, dysmorphic features were noted, with a short neck, frontal bossing, retrognathia, a thin upper lip, a high arched palate, gingival hypertrophy and protruding upper central incisors.

We evaluated motor coordination by using the Movement Assessment Battery for Children [35]. The score was significantly below the mean for both chronological age (CA) and mental age (MA) normative data, except for the Ball Skills subtest, on which she scored slightly below the mean for MA normative data. In particular, she failed to place the coins in the box. Moreover, the expected reduction in reaction times in the second trial was not observed. In particular, she had extreme difficulty performing the task with her left hand. Finally, in the Threading Beads subtest, she showed poor oculo-motor coordination. She showed general difficulties in starting new actions or new tasks and reduced attention times. Sometimes, echolalia occurred. Nevertheless, she displayed good responsiveness and very high relational ability.

The neuroanatomical features were investigated by a brain Magnetic Resonance Imaging (MRI) exam. Possible genetic correlates were also investigated by exome sequencing and whole genome Array-CGH. The neuropsychological and behavioral profile was extensively evaluated and followed-up for 10 years. The assessments were conducted at four time points: At first evaluation, the girl was 5.7 years old; at second evaluation, the girl was 8 years old; at third evaluation, she was 11 years old; at the fourth and last evaluation, occurring 10 years after the first assessment, she was 15 years old.

Concerning ongoing treatments, at first evaluation, she attended speech therapy (continued until 10 years of age), physiotherapy and psychomotricity therapy (twice a week) at a rehabilitation center in her town. At the follow-up evaluations, she attended physiotherapy and psychomotricity (twice a week) as well, and was followed by a teacher aid and an educator at school until 15 years of age. She underwent pharmacological treatment for seizures (valproate and levetiracetam at 11 years of age and oxcarbazepine at 15 years of age). At 15 years of age, the girl started to assume antipsychotic drugs in association with anti-epileptic medications (valproate, oxcarbazepine; clonazepam; risperidone). The neurological status remained stable until the last evaluation at 15 years of age (seizures free), thus no further neuroradiological examination was performed.

## 2. Materials and Methods

### 2.1. Neuroradiological Examination

Magnetization Prepared Rapid Gradient Echo (MPRAGE) T1-weighted images (TR = 11.4 ms, TE = 4.4 ms, flip angle = 15) were obtained with a Siemens Vision Magnetom MR system (Siemens Medical Systems, Erlangen, Germany) operating at 1.5 T; acquisition plane: sagittal. High spatial resolution brain sampling of 0.97 by 0.97 by 1.25 mm, allowing images to be rotated by small angles in the three orthogonal planes, facilitated identification of anatomical landmarks for the selection of the regions of interest. This sequence produced 128 contiguous slices of 1.25 mm thickness, which covered the whole brain.

### 2.2. Genetic Analysis

#### 2.2.1. Exome Sequencing

Genomic DNA was extracted from peripheral blood of the girl and her parents, using commercial kit. Informed consent was obtained from all participating subjects according to the Declaration of Helsinki. Whole exome sequencing was performed using Illumina HiSeq X, and the resulting 150 bp paired-end reads were aligned to the GRCh38 reference genome. Data analysis was performed using an in-house implemented pipeline, mainly based on the Genome Analysis Toolkit (GATK v3.7). To prioritize variants, we applied a sequential filter to retain only those variants with the following characteristics: (a) potential effect on protein and transcript; (b) consistency with the suspected inheritance model (de novo or autosomal recessive); and (c) consistency with a neurodevelopmental/neurological phenotype. The pathogenicity of the identified missense variants were investigated using PolyPhen-2, SIFT, Mutation Assessor and CADD, while conservation of the affected residue was assessed by ClustalW2 (http://www.ebi.ac.uk/Tools/msa/clustalw2/ accessed on 20 January 2022).

#### 2.2.2. Whole Genome Array-CGH

DNA was also analyzed by CGH-microarray using high resolution Affymetrix SNP- array GeneChip 6.0 to exclude potential pathogenic Copy Number Variations (CNVs). Data were analyzed using the Agilent Cytogenomics software (Agilent Technologies, Santa Clara, CA, USA; Agilent Cytogenomics v3.0.6.6).

#### 2.2.3. Homology Modeling

Homology modelling of CUB 1, Sushi 1, CUB 10 and Sushi 10 domains of CSMD1 was based on Protein Data Bank (PDB) structures showing the highest amino acid identity encompassing the identified variants: CUB 1 domain (a.a. 32–140) on PDB 3KQ4 (40% a.a. identity); Sushi 1 (143–203) on PDB 1LY2 (44% a.a. identity); CUB 10 (a.a. 1625–1733) on PDB 5FWS (32% a.a. identity); Sushi 10 (a.a. 1739–1799) on PDB 1H03 (35% a.a. identity). MODELLER software (University of California San Francisco, San Francisco, CA 9, USA; v. 9v17) was used [36]. To represent a mutual arrangement of the contiguous domains for CUB 1/Sushi 1 and CUB 10/Sushi 10 of the CSMD1 protein as observed in experimental structures, the individual modelled domains in each CUB/Sushi pair were superimposed onto the corresponding CUB and Sushi domains of the C1S protein structure (PDB 4LOS), as the latter contains contiguously arranged CUB/Sushi domains. Molecular structures were rendered with PyMOL (http://www.pymol.org accessed on 20 January 2022).

### 2.3. Neuropsychological and Behavioral Examination

The neuropsychological assessment consisted of a battery of tests tailored to the patient’s age, cognitive level and level of cooperation. If the girl performed lower than expected for her CA, the performance was further evaluated considering norms for her MA. We used the Italian version and the Italian reference norms in all tests. Regarding tests for which norms were unavailable, the girl’s scores were directly contrasted with those of two control groups of healthy children using the Crawford and Garthwaite’s procedure [37], respectively, per each evaluation. At first evaluation (5.7 years of age), 1 control group (CA-1-matched) included 12 5-year-old children (6 F) with a mean CA of 5 years and 8 months (SD = 4 mo); the other (MA-1-matched) consisted of 10 3-year-old children (6 F) with a mean MA of 3 years and 5 months (SD = 1 y). At the evaluation occurring at 11 years of age, we considered 1 control group (CA-2-matched), including 12 11-year-old children (6 F) with a mean CA of 11 years and 2 months (SD = 3 months), and a second control group (MA-2-matched) consisting of 12 6-year-old children (6 F) with a mean MA of 6 years and 7 months (SD = 2 months). At the last evaluation, occurring after 10 years from the first evaluation, when the girl was 15 years old, it was not possible to administer the entire battery of tests because she was poorly collaborative. The girl’s nonverbal intelligence score, from the Leiter International Performance Scale-Revised [38], was considered as a measure of her cognitive ability. This decision was supported by the hypothesis that impaired motor and linguistic functions, associated with cerebellar agenesis [5,13,26,39], could affect results on the Wechsler Intelligence Scales. Therefore, the matching criterion for the MA-matched control group was her Leiter-R MA score.

Different neuropsychological domains, adaptive level and psychopathological symptoms were evaluated. Language: lexical expression [40,41], morphosyntactic expression (The Repetition Sentences Task from Language Assessment Test, [42]), lexical comprehension (Peabody Picture Vocabulary Test, [43]) and morphosyntactic comprehension (The Grammar Comprehension Test, [44]). Phonological awareness: syllabic blending and segmentation for the first evaluation and phonological blending and segmentation when the girl was 11 years old (Metaphonologic Competences Test, [45]). Memory: verbal, visual and spatial short-term memory (Word Span Test, Visual Span Test, Spatial Span Test from Promea, [46]); phonological working memory (Nonword Repetition Test, Promea, [46]); verbal, visual and spatial episodic memory (Verbal Learning test, Visual-Object and Visual-Spatial Learning test form Promea, [46]); semantic verbal memory (Categorical Fluency Test, [46]); and procedural learning (Serial Reaction Time Task; [47,48]). Executive functions: selective and sustained visual attention (Bells test “test delle Campanelle”, [49]); planning abilities (Tower of London Test, [50]); inhibition (Go/No-Go Task, [51]). Perceptual and visual-spatial skills: visual-motor integration (Visual Motor Integration Test, [52]); perceptual abilities (Visual Perception Test, [53]). Academic skills: the assessment occurred at 8 and 11 years of age on reading (MT reading battery, [54]), writing [55] and math abilities (AC-MT batteries, [56]). Adaptive level: Vineland Adaptive Behavior Scales (VABS, [57,58]). Psychopathological profile: psychopathological symptoms per each evaluation time (Child Behavior Checklist—CBCL, [59]; Kiddie Schedule for Affective Disorders and Schizophrenia—present and Lifetime Version—K-SADS, [60]).

## 3. Results

### 3.1. Neuroradiological Examination

The brain MRI exam, performed at first evaluation, showed the presence of the vermian lobules I-V on the right and an embryonal formation of vermian lobules I-V on the left. Furthermore, the bilateral superior cerebellar peduncles, part of the left hemisphere lobule VI (which is wider in the right hemisphere), the Crus II and part of the hemisphere lobules VIIB, VIIIA and VIIIB bilaterally appeared preserved (Figure 1).

At the cortical level, the sulcation process did not develop normally. The primary (e.g., right central) and secondary convolutions (bilateral parieto-occipital and frontal sulci, and part of the right temporal gyri) were not precise in their location and/or configuration (Figure 2, Panel A). Numerous small gyri were distributed in the parietal lobes and in the regions of the Sylvian fissure, bilaterally, involving the temporal-frontal lobe and the insulae (Figure 2, Panel B). Both temporal lobes seemed spared, but in the right temporal lobe, the gyri were partially misplaced, with the medial temporal gyrus hiding the superior temporal gyrus (see Figure 2, Panel C).

Moreover, the MRI showed reduced white matter, which, in correspondence with the cerebral cortex abnormalities, appeared relatively thin and with atypical reorganization. Furthermore, no ectopic foci were found in the brain. The ventricular system showed bilateral enlargement of the occipital horns and an alteration of both trygon horns, which was more evident on the right side. Volumetric reduction of the brainstem was present for all components.

### 3.2. Genetic Analysis

The association between microcephaly, polymicrogyria and cerebellar agenesis prompted us to screen for tubulin genes (TUBA1A, TUBA8, TUBB2A, TUBB2B, TUBB3, TUBB4A, TUBB, TUBG1 [33]), which were all negative. Mutations in PTF1A, another gene associated with cerebellar agenesis [10], were also ruled out. CNVs were excluded by array-CGH analysis. Exome sequencing detected compound heterozygous missense variants in CSMD1 (OMIM* 608397) in the patient. The maternal-inherited variant leads to a Glutamine to Lysine change in position 1782 (NM_033225.5:c. 5344C>A; p.Gln1782Lys; rs202157459) with a Minor Allele Frequency/MAF = 0.002 classified as VoUS according to ACMG Standards and Guidelines, while the paternal-inherited variant leads to a Serine to Asparagine change in position 188 (NM_033225.5: c.563G>A; p.Ser188Asn; rs36042022) with an MAF = 0.002 classified as VoUS and is predicted damaging by in silico tools.

#### Homology Modeling

The p.Ser188Asn and Gln1782Lys variants affect the CSMD1 protein in the region characterized by several alternating CUB and Sushi domains. In particular, these two amino acid substitutions involve quite conserved residues in the first and tenth Sushi domains at sites exploited for the intramolecular interactions with contiguous CUB domains (Figure 3). Therefore, it can be expected that the p.Ser188Asn and Gln1782Lys variants introduce defects in the structural packing of the first and tenth Sushi domains with their proximal CUB domains, leading to structural distortions in the overall arrangement of the multiple CUB/Sushi domains and CSMD1 protein malfunctioning.

### 3.3. Neuropsychological and Behavioral Examination

The timeline of the neuropsychological and behavioral evaluation is summarized in Figure 4.

#### 3.3.1. Intellectual Level

At first evaluation, when the girl was 5.7 years old, her nonverbal MA was 3.7 years and short IQ was 73. When she was 8 years old, her nonverbal MA was 4.10 years and her short IQ was 65. When she was 11 years old, the MA was 6.7 years and short IQ was 71. When the girl was 15 years old, her MA was 5.7 years and her short IQ was 52.

#### 3.3.2. Neuropsychological Tasks

A detailed description of the results is reported in the Appendix A. Table 1 shows the girl’s raw scores on neuropsychological tasks for the evaluations at 5.7 and 11 years of age. The girl’s raw scores were compared to the lower limit of the 95% tolerance interval of the relative CA or MA norms (when normative data were available) and the mean score and the standard deviation of the CA- and MA-matched control groups (when normative data were unavailable). Comparisons with MA reference norms or MA-control group data are not reported if the girl’s performance was on average for CA. After normalization, a score was considered pathological if it fell below the fifth percentile for the normative population and slightly below the mean if it was in the range of the fifth–tenth percentile of the normative population. A score was considered in the average range if it was higher than the tenth percentile.

A picture of decline in some neuropsychological functions and a dramatic worsening of adaptive and psychopathological domains emerged. The first neuropsychological evaluation detected impairment in a few areas, namely, lexical expression and comprehension, episodic and semantic verbal memory, planning abilities, sustained visual attention and inhibition, while adaptive and psychopathological domains were preserved. At 11 years of age, she exhibited a worsening of some abilities: The neuropsychological evaluation revealed global impairment in language and phonological awareness, compromised verbal short-term and semantic memory, associated with a global impairment in the executive functions, with the exception of sustained visual attention. The evaluation of her academic skills detected significant impairment in reading, writing and math abilities. At 15 years of age, a global decline in her adaptive abilities emerged, associated with the rise of significant multiple psychological symptoms referred to mood, anxiety, attention, aggressive behavior, conduct/dyscontrol and hyperactivity.

A qualitative summary of the girl’s developmental clinical outcome in the neuropsychological, adaptive and psychopathological domains is available in Table 2.

## 4. Discussion

In this study, we report the case of a girl who came under our observation for documented cerebellar agenesis. A subsequent neuroradiological investigation revealed subtotal cerebellar agenesis and, at the cortical level, bilateral polymicrogyria distributed in the parietal lobes and regions of the Sylvian fissure. Whole exome sequencing identified two rare variants in *CSMD1*, p.Ser188Asn and Gln1782Lys. The neuropsychological and behavioral evaluations and the follow-ups during 10 years showed impairment in multiple cognitive functions in childhood, with a progressive decline in cognitive and adaptive abilities and the emergence of psychopathological symptoms in late adolescence. To our knowledge, this is the first description of the co-occurrence of rare variants of *CSMD1* gene and the rare neurodevelopmental condition of cerebellar agenesis, therefore, we were interested in deeply characterizing the developmental cognitive and behavioral outcome of this patient. The evaluations were carried out at four time points: when the girl was 5.7 years old, 8 years old, 11 years old and 15 years old.

### 4.1. Neuropsychological Outcome

Intellectual disability persisted through six years in the mild severity range, which is in line with existent literature describing the presence of different degree of intellectual disability in cases of cerebellar agenesis and other cerebellar congenital malformations [28,61,62,63]. However, the intellectual disability turned to a moderate severity range in adolescence, as a result of a progressive decline in cognitive and adaptive abilities. Prosody and lexical expression and comprehension deficits emerged since the first evaluation, while morphosyntactic comprehension and expression deficits arose with the time. Language delay is commonly reported in cases of total or partial cerebellar agenesis, although the degree of impairment differs between studies [24,26,39,61,64,65,66].

Considering memory abilities, the girl showed marked deficits on verbal phonological working memory and short-term memory, as generally described in cases of cerebellar agenesis [13,24]. Concerning long-term memory, she showed different degrees of impairment on explicit verbal (episodic and semantic) memory over time, in agreement with literature on cerebellar agenesis or hypoplasia [13,26,64]. Explicit long-term verbal memory deficits have been reported independently of the hypoplastic cerebellar areas (vermian or hemisphere), suggesting that this is a prominent feature associated with cerebellar agenesis [67]. Conversely, spatial long-term memory improved at 11 years of age. This finding could suggest a possible compensation by the medium temporal regions, as a result of developmental brain neuroplasticity [68]. Similarly, implicit long-term memory was found preserved with a typical learning curve. This evidence is slightly in contrast with literature on cerebellar conditions, showing marked deficits in procedural learning tasks. Indeed, the cerebellar contribution in implicit learning has been documented in several studies using implicit motor learning tasks [69,70,71,72]. However, in this study, we did not measure implicit motor leaning but perceptual sequential learning, which is thought to mainly involve subcortical brain regions—such as the hippocampus and basal ganglia [73,74]. It could be speculated that the contribution of extra-cerebellar regions could have supported the relatively preserved performance of the girl in our task.

Concerning executive functions, the girl showed marked and stable impairment over time in several domains. This impairment mainly involved sustained attention, planning and inhibition but not selective visual attention. A great number of omissions characterized her performance on the Go/No-go task, in accordance with previous reports on other cerebellar conditions [75,76]. Overall, our findings are in line with the recognized role of the cerebellum in executive functions [15], in particular in the so-called ***cold*** executive functions. Traditionally, indeed, executive functions can be classified into ***cold*** executive functions (i.e., merely cognitive processes, such as working memory) and ***hot*** executive functions (involving the processing of information related to reward, emotion and motivation) [77]. Despite us being unable to test the girl’s executive functions at 15 years of age because of her reduced compliance to the assessment, the phenotype emerging could be also interpreted at the light of some considerations concerning the role of ***hot*** executive functions. Recent findings indicate that cerebellar inputs to the ventral tegmental area modulate the reward pathway and play a prominent role in social behavior; thus, the cerebellum can regulate functions related to decision making and emotional control [78]. This is consistent with the behavioral changes that we observed when the girl was 15 years old, characterized by behavioral dyscontrol and low motivation for the administered tasks. Conversely, perceptual and visual-spatial abilities were average for her MA. These results are not consistent with a recent case report which documented an impairment of multisensory integration by measuring reaction times after the presentation of visual, auditory and audiovisual stimuli in a patient with cerebellar agenesis without other brain malformations [25]. Definitive conclusions on this aspect cannot be established, considering that available results derive from single-case studies.

In addition, it must be noticed that the girl showed severe impairment on academic skills. Globally, the observed impairment in learning abilities could be linked to cerebellar abnormalities, in line with the hypothesis of cerebellar involvement in reading and writing disabilities [79].

However, the presence of associated brain abnormalities, such as parietal and Silvian fissure polymicrogyria in our case, may have concurred in partially explaining or in exacerbating the neuropsychological outcome we have observed. In particular, polymicrogyria in bilateral perisylvian regions has been associated with a number of language deficits, including lexical production and comprehension [80,81,82,83,84,85], as well as verbal memory [86]. Moreover, fronto-parietal abnormalities have been associated with executive functions, working memory and learning deficits [87,88,89,90].

### 4.2. Behavioral, Psychopathological and Adaptive Outcome

Regarding the psychopathological and adaptive outcome, the girl exhibited an evident decline in global functioning over time. Although initially preserved, at 15 years of age, she displayed marked signs of behavioral dyscontrol and a wide range of psychopathological symptoms. This symptomatology seems in line with the recognized role of the cerebellum as an “emotional pacemaker” [91]. It is important to note that the girl’s low cognitive resources may have negatively affected her psychopathological profile because coping strategies are generally poor and psychopathological risk is high in people with low IQ and ID [92,93,94]. Moreover, some considerations on the possible role of the seizures on the girl’s behavioral phenotype should be introduced, although literature reveals that the impact of specific epilepsy-related characteristics on challenging behavior in people with intellectual disability may be modest [58].

### 4.3. Etiological Considerations

Taken together, deficits in executive functions, language and verbal memory, as well as reduced cognitive resources, a decline in adaptive level and the emergence of overt psychopathology resemble the descriptions of the Cerebellar Cognitive Affective Syndrome—CCAS [95], a condition involving the impairment of executive functions and language, psychological changes and emotional blunting [91,92]. CCAS arises from damage to the “cognitive cerebellum”, localized in the cerebellar posterior lobe, and the “affective cerebellum”, localized in vermal lobules; the clinical manifestation of the syndrome is more severe in cases of diffuse cerebellar lesions [15,96]. The fact that her impairment in the first years of life was less severe than in adolescence is in accordance to reports of delayed onset of psychiatric symptoms in postoperative cases of cerebellar injury [97]. Moreover, the co-occurrence of parietal and Silvian fissure polymicrogyria may have concurred in exacerbating the observed deficits, as reported in the studies on association between cerebellar agenesis and polymicrogyria [86].

Another possible explanation for the dramatic decline in behavioral and adaptive functioning in our case could be ascribed to CSMD1 protein alteration. *CSMD1* has a recognized role in a wide range of cognitive and psychopathological conditions, however, its specific role in neurodevelopment has yet to be clarified. Variants in *CSMD1* have been associated with deleterious effects across a number of neurological and neuropsychiatric phenotypes (see the Appendix A): autism [98], bipolar disorder [99], Alzheimer’s disease [100], Parkinson’s disease [101] and schizophrenia [102,103]. However, little is known about the effects associated with the p.Ser188Asn and Gln1782Lys variants detected in our case.

CSMD1 is expressed in rat brain tissue, especially in the hippocampus, cerebellum, olfactory bulb, spinal cord, thalamus and brain stem [104,105]. CSMD1 transcript is also expressed in the human fetal brain, adult brain and cerebellum, whereas in lymphocytes and fibroblasts, no expression is detected (data not shown, available on request).

*CSMD1* has been described as a regulator of complement activation and inflammation in the developing central nervous system [104]; it was also suggested that it could play an important role in modulating the ratio of dopamine and serotonin in the Cerebrospinal Fluid [106], and it has been detected in the synaptic cleft proteome [107]. Interestingly, CSMD2 and CSMD3, the other members of the CUB and Sushi multiple domain protein family associated with neuropsychiatric disorders, have been found to interact with proteins of the post-synaptic density, and they are required for the maintenance of dendritic spine density or to regulate dendrite development [108,109]. Studies on *CSMD1* knockout mice suggest that the depletion of CSMD1 expression is linked with abnormal emotion/affect behavior, hyperactivity and increased anxiety-related response [105]. The associations of *CSMD1* variations and low abilities in a range of domains have been described, in particular with general cognitive abilities, strategy formation, planning, set shifting and episodic verbal memory [110,111]. Intriguingly, three affected members of a family with learning difficulties, aggressive behavior and facial dysmorphisms associated with epilepsy were found to carry a t(4;8)(p15.2;p23.2) translocation that interrupted the coding sequence of CSMD1 at 8p23.2 [112], while a partial duplication of CSMD1 was associated with developmental delay, autism and myoclonic seizures in a child [113]. Thus, an effect of *CSMD1* variations on cognitive and psychopathological characteristics and development could be hypothesized.

Finally, considering that CSMD1 is a plasma membrane protein of growing neurons [104], an impact of *CSMD1* rare variants on brain development and brain abnormalities cannot be excluded. Indeed, studies have reported association between *CSMD1* genetic variants and brain functions and dysfunctions beyond the neuropsychiatric and cognitive domains. For example, at the brain imaging level, associations have been reported between genetic variants in *CSMD1* and alterations of brain fiber tracts [114] or the default mode network [115]. This suggests that independent variants in the *CSMD1* gene might be implicated in different brain-related phenotypes [111].

In this perspective, the observed phenotype could be interpreted as the result of a complex interplay between cerebellar abnormality, brain malformation and molecular factors, accounting for the general cognitive and psychiatric developmental outcome.

In Figure 5, we have summarized the etiological hypotheses for the observed cognitive and psychiatric phenotype. We have identified some associations already documented, such as the relation between cerebellar agenesis and the cognitive and psychiatric profile, the relation between polymicrogyria and some cognitive characteristics and the relation between *CSMD1* variants and cognitive and psychiatric symptoms. Moreover, we have postulated a potential association between rare *CSMD1* variants and cerebellar agenesis and/or polymicrogyria that needs to be further investigated. Each possibility could be considered individually and in combination.

## 5. Conclusions

A better understanding of the etiopathogenesis and clinical outcome of cerebellar malformations is essential to disentangle the role of cerebellum for controlling and modulating the development of cognition and emotional and adaptive behavior. To our knowledge, this is the first reported case of cerebellar agenesis and cortical abnormalities associated with rare variants of *CSMD1* with a documented neuroradiological and extensive, longitudinal neuropsychological and behavioral evaluations. The conclusion derived from the developmental observation of the present case allows us to trace an evolutionary trajectory that can be crucial to guide clinicians in the diagnosis and management of these rare co-occurrences. Although the role of extra-cerebellar pathology in behavioral consequences of cerebellar condition is still poorly understood [116], the possible co-occurrence of extra-cerebellar abnormalities, as in our case, should be taken into account while setting up rehabilitative pathways, for example, parent training, cognitive behavioral therapy and/or applied behavioral analysis to treat attentive dysregulation and disruptive behaviors, as well as medication such as antipsychotics and mood stabilizers to reduce aggressive behaviors and mood dysregulation. Altogether, this report suggests the importance of a transdisciplinary neurobehavioral approach that takes into account genetic factors and behavioral and cognitive symptoms for patients with a complex phenotype. However, larger-scale studies are required for a better understanding of the multi-dimensional complexity of the genetic and epigenetic mechanisms that account for the inter-individual variability in brain function and development.

## Figures and Tables

**Figure 1 ijerph-19-01224-f001:**
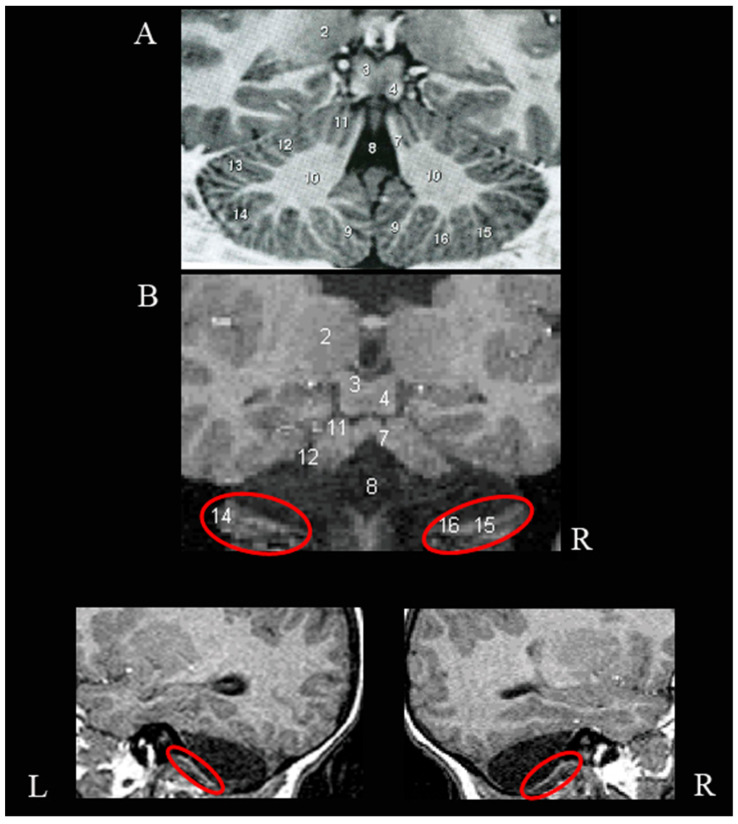
Agenesis of the cerebellum. Neuroradiological examination of the cerebellum when the girl was 5.7 years old. (Panel **A**) is a slice of the cerebellum atlas, (Panel **B**) is the same slice in the girl’s cerebellum. Cerebellar regions are named according to Schmahmann et al. (1999): 2, pulvinar thalami; 3, superior colliculus; 4, inferior colliculus, 7, superior cerebellar peduncle; 8, fourth ventricle; 11, hemisphere lobules IV-V (anterior quadrangular lobule); 12, hemisphere lobule VI (posterior quadrangular lobule); 14, Crus II (inferior semilunar lobule); 15, hemisphere lobule VIIB (gracilis lobule); 16, hemisphere lobules VIIIA and VIIIB (biventer lobule). L = left; R = right.

**Figure 2 ijerph-19-01224-f002:**
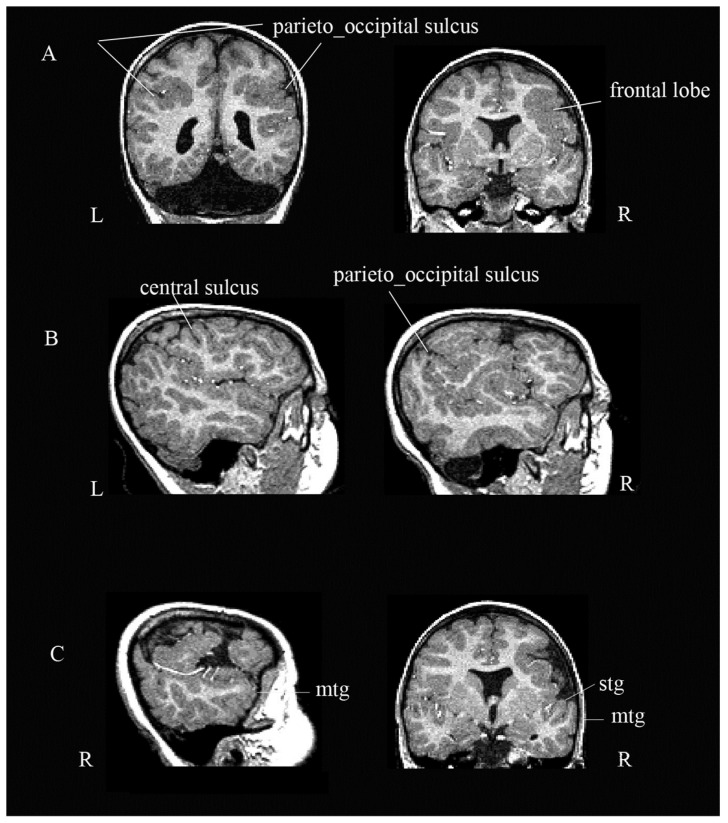
Cortical abnormalities. The figure shows the abnormal sulcation process with mis-location and mis-configuration of sulci and gyri. (Panel **A**) shows abnormalities in both parietal lobes and the right frontal lobe. (Panel **B**) shows polymicrogyria in the parietal lobes and the regions of the sylvian fissure bilaterally. (Panel **C**) shows lateral and coronal views of the right temporal lobe; the medium temporal gyrus (mtg) is hiding the superior temporal gyrus (stg). L = left; R = right.

**Figure 3 ijerph-19-01224-f003:**
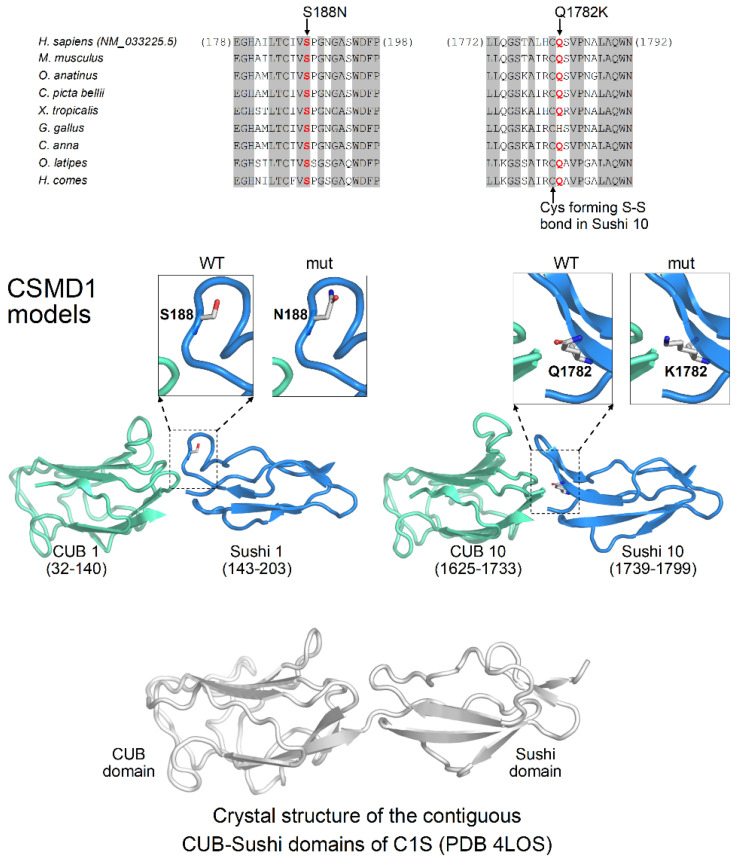
Homology Modeling of the *CSMD1* variants. Top: multiple sequence alignment of CSMD1 protein among organisms around the sites of p.Ser188Asn and Gln1782Lys mutations. Invariant columns are grayed. Bottom: molecular models of the Sushi domains (blue ribbons) involved by mutations. The CUB domains (green ribbons) immediately N-terminal are also modeled, and both Sushi and CUB domains are mutually positioned as in the PDB structure 4LOS (white ribbons), representing the crystal structure of the contiguous CUB/Sushi domains of complement C1s subcomponent.

**Figure 4 ijerph-19-01224-f004:**
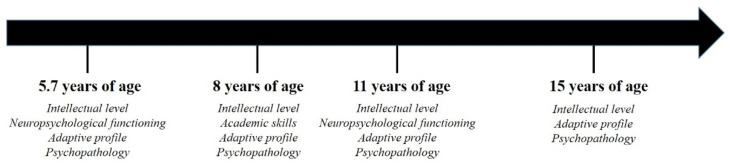
Timeline of the neuropsychological and behavioral evaluations.

**Figure 5 ijerph-19-01224-f005:**
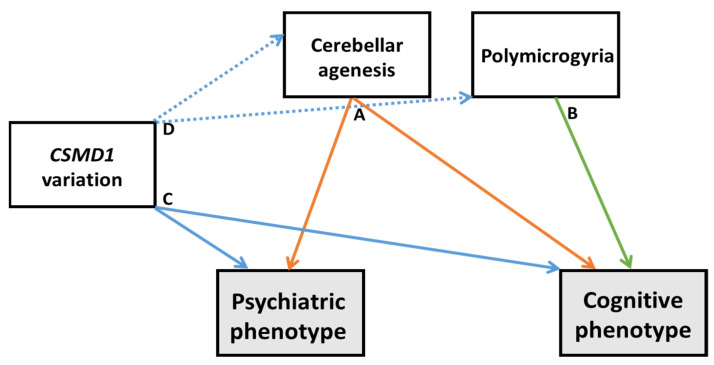
Hypotheses for the observed phenotype. Continuous lines indicate documented relationship between neuroanatomical, molecular and psychiatric/cognitive characteristics. In particular, option A indicates the contribution of cerebellar agenesis on the psychiatric and cognitive phenotypes; option B highlights the role of polymicrogyria in the observed cognitive profile; option C underlines the direct role of *CSMD1* in the psychiatric and cognitive phenotypes. Dotted lines indicate a putative relationship between *CSMD1* variations and the observed neuroanatomical features (option D).

**Table 1 ijerph-19-01224-t001:** Raw scores on neuropsychological tests obtained by the girl at 5.7 years of age and 11 years of age. For each test (depending on the availability of normative data), the lower limit of the 95% tolerance interval for the normative population, or mean scores and standard deviations (*) of chronological-age (CA-1-matched) and mental-age (MA-1-matched) matched control groups at first evaluation and chronological-age (CA-2-matched) and mental-age (MA-2-matched) matched control groups at second evaluation, are reported. Letters “a”, “aa”, “b” and “bb” indicate if the girl scored below average.

Neuropsychological Assessment	First Evaluation (5.7 yrs)	Lower Limit Of 95% Tolerance Interval For Chronological Age Norms (5.8)/*CA-1 Matched Controls (N = 12), Mean (SD)	Lower Limit of 95% Tolerance Interval for Mental Age Norms (3.7)/*MA-1 Matched Controls (n = 10), Mean (SD)	Second Evaluation (11 yrs)	Lower Limit of 95% Tolerance Interval for Chronological Age Norms (11.7)/*CA-2 Matched Controls (n = 12), Mean (SD)	Lower Limit of 95% Tolerance Interval for Mental Age Norms (6.7)/*MA-2 Matched Controls (n = 12), Mean (SD)
**Language**						
*Expression*						
Lexical	9 ^aa,bb^	15.06	10.88	27 ^aa^	31.8	12
Morphosyntactic	7.5 ^aa^	10.42	3.46	10 ^aa,bb^	N.A.	11.9
*Comprehension*						
Lexical	27 ^aa,b^	52	11	84 ^aa,b^	117	84
Morphosyntactic	67.2	33		43.9 ^aa,b^	N.A.	37.9
**Phonological Awareness**						
Syllabic Blending	12	11				
Syllabic Segmentation	14	8				
Phonological Blending				0 ^aa,bb^	N.A.	5
Phonological Segmentation				0 ^aa,bb^	N.A.	0
**Memory**						
*Short-term and Working Memory*						
Word Span	3	2.6		3 ^aa,b^	3.6	2.8
Nonword Repetition	7 ^aa,bb^	13	*26.6 (9.5)	28 ^a^	28	20
Visual Span	0.4 ^aa^	2.6	*1.82 (1.23)	3 ^a^	3	2.6
Spatial Span	2.8	2.6		4.2	3	
*Explicit Long-term Memory*						
Word Recall Immediate	11	7		14 ^aa^	19	11
Word Recall Delayed	0 ^aa,bb^	1	*3.8 (1.5)	4 ^aa^	6	2
Semantic	7 ^aa,bb^	14	*19.9 (4.6)	18 ^aa,b^	33	18
Visual Immediate	15	5		24 ^a^	24	13
Visual Delayed	7	2		9 ^a^	9	4
Spatial Immediate	13 ^a^	8	*26.3 (10.3)	44	22	
Spatial Delayed	0 ^aa,b^	1	*7.5 (3.4)	15	9	
*Implicit Long-term Memory*						
SRTT I (random)	840	*829 (171)		641	*471 (61)	
SRTT II (ordered)	820	*659 (167)		688	*441 (42)	
SRTT III (ordered)	766	*579 (118)		605	*417 (71)	
SRTT IV (ordered)	734	*528 (156)		716	*405 (77)	
SRTT V (random)	781	*727 (161)		855	*442 (55)	
**Executive Functions**						
*Attention*						
Selective	19 ^aa^	21.8	11.6	26 ^aa,b^	43	24.9
Sustained	46 ^aa^	65.8	41.1	96 ^aa^	119.9	78.5
*Planning*						
TOL	3 ^aa,bb^	16	13	15 ^aa,bb^	23	18
*Inhibition*						
Go RTs	615	*455 (103)	*612 (236)	570 ^aa^	*223 (98)	*453 (101)
Go omissions	1	*0.5 (1.24)	*1.2 (1.2)	0	*0.3 (0.2)	*0.8 (1.2)
NoGo RTs	801	*701 (185)	*843 (151)	625 ^aa^	*328 (122)	*612 (82)
NoGo errors	2	*3.66 (6.82)	*3.8 (4.2)	6 ^aa,bb^	*1.2 (0.6)	*2.3 (1.3)
NoGo omissions	33 ^aa,bb^	*1.25 (2.1)	*3.8 (3.1)	1	*0.4 (1.1)	*5.3 (4.1)
**Visual-Spatial Abilities**						
*Visual-motor integration*						
Integration	6 ^aa^	8	2	12 ^aa^	14	9
Visual perception	16	9		13 ^aa^	16	10
Motor coordination	9 ^a^	9	4	13 ^aa^	16	9
*Perceptual abilities*						
Spatial Positions	5	4		15 ^aa^	16	7
Confounding Background	6	5		16	9	

^a^ Slightly below CA healthy children. ^aa^ Significantly poorer than CA healthy children. ^b^ Slightly below MA healthy children. ^bb^ Significantly poorer than MA healthy children. RTs = reaction times (milliseconds); SD = standard deviation.

**Table 2 ijerph-19-01224-t002:** Summary of the girl’s developmental clinical outcome in the neuropsychological, adaptive and psychopathological domains. For neuropsychological and adaptive domains, the dark green indicates preserved abilities for chronological age, whereas light green indicates preserved abilities for mental age; finally, red indicates impaired abilities for both chronological and mental age. For the psychopathological domain, light green indicates symptoms or traits not fully meeting criteria for a diagnosis (subthreshold symptoms) based on the clinical judgement, supported by K-SADS interview and CBCL results.

Neuropsychological and Behavioural Assessment	5 ys	8 ys	11 ys	15 ys
**Neuropsychological Measures**				
**Language**	Lexical expression				
	Morphosyntactic expression				
	Lexical comprehension				
	Morphosyntactic comprehension				
**Phonogical awareness**	Syllabic blending				
Syllabic segmentation				
**Memory**	Short-term verbal				
Short-term visual				
Short-term spatial				
Phonological working memory				
Episodic verbal memory (immediate)				
Episodic verbal memory (delayed)				
Semantic verbal memory				
Episodic visual memory (immediate)				
Episodic visual memory (delayed)				
Episodic spatial memory (immediate)				
Episodic spatial memory (delayed)				
Procedural learning				
**Executive**	Selective visual attention				
	Sustained visual attention				
	Planning abilities				
	Inhibition				
**Perceptual/visual-spatial**	Visual-motor integration				
Perceptual abilities				
**Academic abilities**				
	Reading				
Writing				
Math				
**Adaptive level**				
	Communication				
	Daily living skills				
	Socialization domain				
	Motor abilities				
**Psychopathological evaluation**				
	Mood				
Anxious/fobia				
Attention				
Aggressive behavior				
PTSD				
Obsessive				
Social problems				
Conduct/dyscontrol				
Hyperactivity				

## Data Availability

The data presented in this study are available on request from the corresponding author.

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
