# Peer review of "Cerebellar Agenesis and Bilateral Polimicrogyria Associated with Rare Variants of CUB and Sushi Multiple Domains 1 Gene (CSMD1): A Longitudinal Neuropsychological and Neuroradiological Case Study"

_ijerph, 2022, doi:10.3390/ijerph19031224_

Round 1
Reviewer 1 Report
Summary: This study is a case-report of a five-years girl with cerebellar agenesis associated with two rare variants in CSMD1. She was followed-up for ten years at four time points with neuroradiological, molecular, neuropsychological (cognition, emotion and behavior) and adaptative examinations.
The article is clear and presented in a well-structured manner. The sections are well-developed, although the introduction could be improved with a more comprehensive review of the literature, since there are few published studies of cerebellar agenesis.
Title: Consider including the term case report in the title.
Methodology and results are clearly explained and the discussion connect the theory to the data. It would be interesting to comment on the changes that appeared at fifteen years of age in light of the neurodevelopmental pattern of cold / hot executive functions and social cognition.
The article is well-written and easy to understand, but I have some minor comments:
Line 76: repeated reference
Line 29: clarify acronyms here and not on lines 197,198
Line 133: eliminate the girl's name initials
Line 441: review the reference (surnames instaed of numbers)
Author Response
Dear Reviewer,
Thank you for your kind letter on Ms. Ref. ijerph-1482551, titled “Cerebellar Agenesis and Bilateral Polimicrogyria associated with rare variants of CUB and Sushi multiple domains 1 gene (CSMD1): A Longitudinal Neuropsychological and Neuroradiological study”.
We are very grateful to your and the reviewers' comments and thoughtful suggestions. We were pleased to have an opportunity to revise and resubmitted this manuscript.
Here are our responses to the reviewer's comments and list of corrections.
Summary: This study is a case-report of a five-years girl with cerebellar agenesis associated with two rare variants in CSMD1. She was followed-up for ten years at four time points with neuroradiological, molecular, neuropsychological (cognition, emotion and behavior) and adaptative examinations.
- The article is clear and presented in a well-structured manner. The sections are well-developed, although the introduction could be improved with a more comprehensive review of the literature, since there are few published studies of cerebellar agenesis.
- We have enriched the introduction with a more comprehensive literature review. In the text (pp.2-3):
“The idea that cerebellar agenesis is entirely symptom free - thus allowing a “normal life” – has largely spread out over time. Starting from one of the first cases reported in lit-erature, dating back to 1940 [16], a growing amount of case descriptions have presented a mild impaired picture associated with cerebellar agenesis. The cases of a 6-year-old girl and of a 58-year-old woman with near total cerebellar agenesis without intellectual disability or neurobehavioral symptoms have been described [17,18]. Moreover, the case of a 22-year-old man [19] with only motor impairment, for instance ataxia, associated with typical neuropsychological development, was observed. However, Ashraf and colleagues [20] reported a case in which cerebellar agenesis was not associated with motor impairment, dysarthria nor nystagmus, but only with learning difficulties. Yu and colleagues [21] described the case of a 24-years-old woman with complete cerebellar agenesis exhibiting only mild to moderate signs of motor impairment and cerebellar dysarthria. More recently, Wu and colleagues [22] reported the case of a 26-year-old patient with complete primary cerebellar agenesis, exhibiting mild to moderate motor impairment associated with impairment in associative motor learning. In other cases, mild intellectual disability with language difficulties but preserved abilities, such as reading or riding a bicycle, as well as adequate levels of affective behavior were detected. An explanation of these results is the hypothesis that the extra-cerebellar motor system can compensate for lost cerebellar motor functions [23].
On the other side, several studies reported histories of patients with multiple and severe deficits associated with cerebellar agenesis such as executive functions, behavioral and/or neuropsychological alterations, multisensory integration associated with ataxic gait, and oculomotor disorders [13,25,26]. Chedda, Sherman and Schmahmann [24] re-ported two cases of children with near-complete agenesis with gross and fine motor deficits, such us oral motor apraxia, impaired saccades and vestibulo-ocular reflex cancellation, clumsiness and mild ataxia. Behavioral characteristics included autistic-like stereotypical performance, obsessive rituals and difficulty in understanding social cues. The most important neuropsychological deficits affected executive functions (perseveration, disinhibition, abstract reasoning, working memory and verbal fluency), visual-spatial abilities (perceptual organization, visual-spatial copying and recall) and expressive language (delay). The severity and range of the motor, cognitive and psychiatric impairments were also related to the extension of the agenesis.
Timmann and colleagues [13] reported the case of a 59-year-old patient, with almost total cerebellar agenesis, with a number of oculomotor, speech and gait control deficits as well as developmental delay and learning deficits. Mild to moderate deficits in IQ and reduced planning abilities, visual-spatial abilities, visual memory and attention or deficits in speech comprehension, verbal learning and declarative memory, were also described [26–28]. More recently, it has been reported the case of a 48 years old man with cerebellar agenesis, exhibiting deficits in executive functions (planning, flexibility and focused attention) and multisensory integration, associated with ataxic gait, and oculomotor disorders [25].
Although documented detailed neurologic, neuropsychiatric, and neuroimaging findings in living patients with total cerebellar agenesis are limited, the patients described present with a variety of developmental motor, cognitive and behavioral abnormalities.”
- Title: Consider including the term case report in the title.
- We have modified the title according to your suggestion as follows: “Cerebellar Agenesis and Bilateral Polimicrogyria Associated with Rare Variants of CUB and Sushi multiple domains 1 gene (CSMD1): A Longitudinal Neuropsychological and Neuroradiological Case Study”. Thank you.
- Methodology and results are clearly explained and the discussion connect the theory to the data. It would be interesting to comment on the changes that appeared at fifteen years of age in light of the neurodevelopmental pattern of cold / hot executive functions and social cognition.
- Unfortunately, we were not able to test the girl’s executive functions at 15 years of age due to the dramatically reduced levels of compliance to the assessment. However, some considerations about the role of hot and cold executive functions could be pointed out. In the text (p.16):
“Concerning executive functions, the girl showed marked and stable impairment over time in several domains. This impairment mainly involved sustained attention, planning and inhibition, but not selective visual attention. A great number of omissions characterized her performance on the Go/No-go task, in accordance with previous reports on other cerebellar conditions [74,75]. Overall, our findings are in line with the recognized role of the cerebellum in executive functions [15], in particular in the so-called cold executive functions. Traditionally, indeed, the executive functions could be classified into cold executive functions (i.e., merely cognitive processes, such as working memory) and hot executive functions (involving the pro-cessing of information related to reward, emotion, and motivation) [76]. Despite we were unable to test the girl’s executive functions at 15 years of age because of her reduced compliance to the assessment, the phenotype emerging could be also interpreted at the light of some considerations concerning the role of hot executive functions. Recent findings indicate that cerebellar inputs to the ventral tegmental area modulate the reward pathway and play a prominent role in social behaviour; thus, the cerebellum can regulate functions related to decision-making and emotional control [77]. This is consistent with the behavioural changes that we observed when the girl was 15 years old, characterized by behavioural dyscontrol and low motivation for the administered tasks.
The article is well-written and easy to understand, but I have some minor comments:
- Line 76: repeated reference.
- We have corrected the mistake and updated the reference section. Thank you.
- Line 29: clarify acronyms here and not on lines 197,198.
- We apologize for the error. We have clarified the acronyms in line 129.
- Line 133: eliminate the girl's name initials
- We removed the initials from the manuscript. Thank you for your comment.
- Line 441: review the reference (surnames instaed of numbers)
- We have revised the reference in line 440. We apologize for the mistake.
Thank you for taking the time and energy to help us improve the paper.
Sincerely Yours,
Deny Menghini and co-authors
Correspondence:
Deny Menghini, PhD
Child Neuropsychiatry Unit
Department of Neuroscience
I.R.C.C.S. Children’s Hospital Bambino Gesù
Viale Baldelli 41, I-00146, Rome (Italy)
e-mail: deny.menghini@opbg.net

Reviewer 2 Report
In this manuscript by Costanzo et al, the authors present a study on CSMD1, a gene associated with neurocognitive and psychiatric alterations, combining genetic and clinical approaches. Overall, the manuscript is well-written, and I recommend it for publication after the authors elucidate the points raised below and revise the manuscript accordingly.
1. Part 2.2.3. Homology modeling: Why did the authors choose the PDB structures 3KQ4, 1LY2, 5FWS, and 1H03 as templates for homology modeling? According to the Protein Data Bank, there are multiple PDB structures available for the UniProt identifier (ID) that corresponds to each of these structures, some even with better resolutions (for example, the resolution of 3KQ4 is 3.3 Angstroms, too low in comparison to that of 6GJE, another structure of the same UniProt ID O60494 but with a much higher resolution). The authors need to briefly explain their choices of PDB templates in this part.
2. Part 3.2.1. Homology modeling: Have the authors checked the structural quality of the homology structures to make sure that there are no geometric anomalies? For example: Is there any issue with the phi-psi dihedral angles, bond angles, bond lengths? Are there atom clashes in the structures? These may be issues with artificially constructed protein models.
3. Some minor remarks:
- The title of the paper: the authors need to change some letters to the capital form to conform with the title case that is used here (e.g. in "Associated", "Rare", "Variants")
- Line 33: "a 5-year old" (not "a 5 years old")
- Line 55: "ageneses are", or "agenesis is" (not "agenesis are")
Author Response
Dear Reviewer,
Thank you for your kind letter on Ms. Ref. ijerph-1482551, titled “Cerebellar Agenesis and Bilateral Polimicrogyria associated with rare variants of CUB and Sushi multiple domains 1 gene (CSMD1): A Longitudinal Neuropsychological and Neuroradiological study”.
We are very grateful to your and the reviewers' comments and thoughtful suggestions. We were pleased to have an opportunity to revise and resubmitted this manuscript.
Here are our responses to the reviewer's comments and list of corrections.
In this manuscript by Costanzo et al, the authors present a study on CSMD1, a gene associated with neurocognitive and psychiatric alterations, combining genetic and clinical approaches. Overall, the manuscript is well-written, and I recommend it for publication after the authors elucidate the points raised below and revise the manuscript accordingly.
- Part 2.2.3. Homology modeling: Why did the authors choose the PDB structures 3KQ4, 1LY2, 5FWS, and 1H03 as templates for homology modeling? According to the Protein Data Bank, there are multiple PDB structures available for the UniProt identifier (ID) that corresponds to each of these structures, some even with better resolutions (for example, the resolution of 3KQ4 is 3.3 Angstroms, too low in comparison to that of 6GJE, another structure of the same UniProt ID O60494 but with a much higher resolution). The authors need to briefly explain their choices of PDB templates in this part.
- Thank you for your comment. When the models were made, the criterion for choosing templates was to find structures sharing the highest amino acid identity with each CUB or Sushi domain of CSMD1 encompassing the discussed mutations. PDB 6GJE is not suitable because it does not encompass the range of interest, which instead did PDB 3KQ4. Here are the a.a. ranges of the two structures:
PDB 3KQ4 encompasses Cubilin a.a. 932-1388
PDB 6GJE encompasses Cubilin a.a. 40-121
Below you can see the pairwise sequence alignments of the particular CSMD1 CUB or Sushi domain and of the PDB structures employed as templates:
CUB 1 domain (a.a. 32-140) on PDB 3KQ4 (a.a. 932-1042; 40% a.a. identity)
CSMD1_32-140 CGGLVQGPNGTIESPGFPHGYPNYANCTWIIITGERNRIQLSFHTFALEENF----DILS
3KQ4_932-1042 CGEILTESTGTIQSPGHPNVYPHGINCTWHILVQPNHLIHLMFETFHLEFHYNCTNDYLE
** :: ..***:***.*: **: **** *:. .: *:* *.** ** :: * *.
CSMD1_32-140 VYDGQPQQGNLKVRLSGFQLPSSIVSTGSILTLWFTTDFAVSAQGFKALYEVL
3KQ4_932-1042 VYDTDSETS--LGRYCGKSIPPSLTSSGNSLMLVFVTDSDLAYEGFLINYEAI
*** :.: . * .* .:*.*:.*:*. * * *.** :: :** **.:
Sushi 1 (143-203) on PDB 1LY2 (a.a. 1-63; 44% a.a. identity);
CSMD1_143-203 HTCGNPGEILKGVL--HGTRFNIGDKIRYSCLPGYILEGHAILTCIVSPGNGASWDFPAP
1LY2_1-63 ASCGSPPPILNGRISYYSTPIAVGTVIRYSCSGTFRLIGEKSLLCITKDKVDGTWDKPAP
:**.* **:* : :.* : :* ***** : * *. * **.. ..:** ***
CSMD1_143-203 FCR
1LY2_1-63 KCE
*.
CUB 10 (a.a. 1625-1733) on PDB 5FWS (a.a. 214-321; 32% a.a. identity);
CSMD1_1625-1733 CGGQYTGSEGVVLSPNYPHNYTAGQICLYSITVPKEFVVFGQFAYFQTA-LNDLAELFDG
5FWS_214-321 CGGNYSAMSSVVYSPDFPDTYATGRVCYWTIRVPGASHIHFSFPLFDIRDSADMVELLDG
***:*:. ..** **::*..*::*::* ::* ** :. .*. *: *:.**:**
CSMD1_1625-1733 THAQARLLSSLSGSHSGETLPLA-TSNQILLRFSAKSGASARGFHFVYQAV
5FWS_214-321 Y--THRVLARFHGRSR-PPLSFNVSLDFVILYFFSDRINQAQGFAVLYQAV
*:*: : * .*.: : : ::* * :. .*:** .:****
Sushi 10 (a.a. 1739-1799) on PDB 1H03 (a.a. 5-65; 35% a.a. identity).
CSMD1_1739-1799 TQCSSVPEPRYGR-RIGSEFSAGSIVRFECNPGYLLQGSTALHCQSVPNALAQWNDTIPS
1H03_5-65 KSCPNPGEIRNGQIDVPGGILFGATISFSCNTGYKLFGSTSSFCL-ISGSSVQWSDPLPE
..*.. * * *: : . : *: : *.**.** * ***: .* :..: .**.*.:*.
CSMD1_1739-1799 CV
1H03_5-65 CR
*
We have integrated the text with details as follows (p.5): “Homology modelling of CUB 1, Sushi 1, CUB 10, and Sushi 10 domains of CSMD1 was based on Protein Data Bank (PDB) structures showing the highest amino acid identity en-compassing the identified variants”.
- Part 3.2.1. Homology modeling: Have the authors checked the structural quality of the homology structures to make sure that there are no geometric anomalies? For example: Is there any issue with the phi-psi dihedral angles, bond angles, bond lengths? Are there atom clashes in the structures? These may be issues with artificially constructed protein models.
- Thank you for your comment. We have checked the structural quality of each model with MolProbity and provided a detailed report of Ramachandran Favoured, Ramachandran Outliers, Rotamer Outliers, C-Beta Deviations, Bad Bonds, Bad Angles, and Clash Scores, which appear to be relatively modest. We have also included the pairwise sequence alignments employed for each homology model with the amino acid similarity annotations (clustal symbols) to allow to further evaluating the reliability of the correspondences of the peptide regions of models and templates. Regarding the assessment of the homology models, here are angle and clash reports by MolProbity (from SWISS-MODEL):
MolProbity assessment of model CSMD1 a.a. 32-140
Ramachandran Favoured 87.85%; Ramachandran Outliers 3.74%; Rotamer Outliers 10.87%; C-Beta Deviations 9; Bad Bonds 13 / 870; Bad Angles 32/1186; Clash Score 18.65.
MolProbity assessment of model CSMD1 a.a. 143-203
Ramachandran Favoured 94.92%; Ramachandran Outliers 1.69%; Rotamer Outliers 8.16%; C-Beta Deviations 3; Bad Bonds 3/477; Bad Angles 15/649; Clash Score 9.91.
MolProbity assessment of model CSMD1 a.a. 1625-1733
Ramachandran Favoured 92.52%; Ramachandran Outliers 0.93%; Rotamer Outliers 6.90%; C-Beta Deviations 5; Bad Bonds 11/849; Bad Angles 19/1154; Clash Score 12.35.
MolProbity assessment of model CSMD1 a.a. 1739-1799
Ramachandran Favoured 96.61%; Ramachandran Outliers 0.00%; Rotamer Outliers 13.46%; C-Beta Deviations 1; Bad Bonds 7/474; Bad Angles 14/647; Clash Score 4.45.
- Some minor remarks:
- The title of the paper: the authors need to change some letters to the capital form to conform with the title case that is used here (e.g. in "Associated", "Rare", "Variants")
- Thank you for your suggestion. We changed some letters to the capital form.
- Line 33: "a 5-year old" (not "a 5 years old")
- We have modified the text accordingly with your correction. Thank you.
- Line 55: "ageneses are", or "agenesis is" (not "agenesis are")
- We have modified the text accordingly with your correction. Thanks.
Thank you for taking the time and energy to help us improve the paper.
Sincerely Yours,
Deny Menghini and co-authors
Correspondence:
Deny Menghini, PhD
Child Neuropsychiatry Unit
Department of Neuroscience
I.R.C.C.S. Children’s Hospital Bambino Gesù
Viale Baldelli 41, I-00146, Rome (Italy)
e-mail: deny.menghini@opbg.net

Round 2
Reviewer 2 Report
In this revised version of the manuscript by Costanzo et al and in their responses to reviewers, the authors have addressed all concerns raised in my previous comments. I hereby recommend the manuscript for publication.